# Effects of β-Mannanase Alone or Combined with Multi-Carbohydrase Complex in Corn–Soybean Meal Diets on Nutrient Metabolism and Gut Health of Growing Pigs

**DOI:** 10.3390/ani14233457

**Published:** 2024-11-29

**Authors:** Gabriela Miotto Galli, Ines Andretta, Camila Lopes Carvalho, Thais B. Stefanello, Bruna Souza de Lima Cony, Alícia Zem Fraga, Karine Ludwig Takeuti, Aline Beatriz da Rosa, Marcos Kipper

**Affiliations:** 1Faculdade de Agronomia, Universidade Federal do Rio Grande do Sul, Porto Alegre 91540000, RS, Brazil; gabi-gmg@hotmail.com (G.M.G.); camila.lps.carvalho@gmail.com (C.L.C.); thaisstefanello@gmail.com (T.B.S.); brunacony@hotmail.com (B.S.d.L.C.); aliciafraga@outlook.com.br (A.Z.F.); 2Departamento de Medicina Veterinária, Universidade Feevale, Novo Hamburgo 93525075, RS, Brazil; karinelt87@yahoo.com.br (K.L.T.); alinebia97@gmail.com (A.B.d.R.); 3Elanco Animal Health, São Paulo 04703002, SP, Brazil; marcos.kipper@elancoah.com

**Keywords:** arabinofuranosidases, β-glucanase, β-mannans, digestibility, enzymes, feed additives, intestinal health, swine, xylanases

## Abstract

**Simple Summary:**

Non-starch polysaccharides (NSP) can reduce performance in pigs due to their deleterious effects on the gastrointestinal tracts of these animals. Corn and soybean meal are the most common ingredients in pig diets, and they contain 6–17% NSP. Soy also contains an amount of β-mannans, which can cause unnecessary and chronic immune stimulation in pigs. This study evaluated whether adding β-mannanase alone (BM) or in combination with a multi-carbohydrase complex (BM + MCC) improves diet digestibility, nutrient and energy metabolism, and gut health in growing pigs. The results showed that the addition of BM and BM + MCC significantly improved dry matter, protein, and energy digestibility, as well as protein and energy metabolizability, compared to the control. Higher nitrogen retention and lower fecal energy were observed in the BM and BM + MCC groups. There was a significant reduction in manure production and fecal moisture with the addition of the enzymes. BM supplementation increased the villus area and villus height-to-crypt depth ratio. Both the BM and BM + MCC diets reduced the fecal calprotectin levels. In conclusion, the use of β-mannanase alone or in combination with a multi-carbohydrase complex improved the nutritional digestibility, nutrient and energy metabolism, and gut health in growing pigs, and they also contributed to reducing manure production in the test animals.

**Abstract:**

(1) Background: This study was performed to evaluate whether the addition of β-mannanase alone or combined with a multi-carbohydrase complex can improve diet digestibility, nutrient and energy metabolism, and the gut health of growing pigs. (2) Methods: Twenty-four pigs (35.56 ± 3.81 kg) were fed a control corn–soybean meal-based diet (no addition) or a control diet with β-mannanase (BM; 300 g/ton) or control diet β-mannanase plus a multi-carbohydrase complex including xylanase, β-glucanase, and arabinofuranosidases (BM + MCC; 300 + 50 g/ton) for 13 days. Total fecal and urine samples were collected from days 6 to 12. The feces samples were collected from all the pigs to determine fecal biomarkers using commercial ELISA tests. Blood samples were collected from all the pigs on day 13 to assess the serum concentrations of acute-phase proteins. All the pigs were euthanized on day 13 for intestinal tissue collection for morphometric analysis. Data were submitted to variance analysis and differences were considered significant at *p* ≤ 0.05 and a trend for 0.05 < *p* ≤ 0.10. (3) Results: The addition of BM and BM + MCC resulted in greater dry matter, protein, and energy digestibility coefficients, and protein (2.87% and 2.60%) and energy (2.61% and 1.44%) metabolizability coefficients compared to control (*p* < 0.05). A greater retention of nitrogen ratio and lower fecal energy were observed in BM and BM + MCC than in the control (*p* < 0.01). Furthermore, the addition of BM and BM + MCC resulted in lower manure production (29.78 and 49.77%, respectively) and fecal moisture (*p* < 0.001) compared to the control. The BM addition resulted in a greater villus area and villi height to crypt depth ratio compared to the control *(p* < 0.05). The addition of BM and BM + MCC diets also reduced the fecal calprotectin levels by 52 and 56% in relation to the control pigs. (4) Conclusions: The use of β-mannanase alone or associated with multi-carbohydrase complex improved nutritional digestibility, nutrient and energy metabolism, and gut health, and reduced the manure production of growing pigs.

## 1. Introduction

Dietary levels above 5% of non-starch polysaccharides (NSP) can reduce pig performance due to deleterious effects (such as greater digesta viscosity and lower digestibility) in the gastrointestinal tract of these animals [1]. Corn and soybean meal are the most common ingredients in pig diets but contain 6 to 17% of NSP [2]. Furthermore, higher concentrations of NSP may reduce total tract digestibility while increasing endogenous amino acid loss [3], mucosal cell turnover rate, mucin secretion, and undigested content [4] in pigs. Thus, several enzymatic addition programs have already been proposed to help animals deal with these negative effects.

Dietary β-mannanase addition can hydrolyze β-mannans [5], which is related to 15 to 35% of total soy NSP [6], and reduce the innate immune response induced by feeding. This enzyme has the potential to improve animal performance, increase apparent metabolizable energy for nitrogen balance, and the true ileal digestibility coefficient of amino acids [7]. It may also have anti-inflammatory and energy-saving properties [8], lowering the cost of engaging the immune system [9].

Carbohydrases such as xylanases, β-glucanase, and arabinofuranosidases are frequently considered in this scenario. These enzymes act on the xylan skeleton, breaking the β-1,4-glycosidic bonds [10] and releasing oligosaccharides, disaccharides, and monomeric pentose sugars, such as xylose and arabinose. These effects were associated with greater growth performance and total apparent digestibility of crude protein, ether extract, calcium, and phosphorus [11], as well as benefits on gut health [12] and minimized fecal emissions [13].

Pigs do not have endogenous enzymes capable of breaking β-1,4-mannosyl bonds, and α-1,6-galactosyl bonds which are found in mannan and xylan molecules. Therefore, these carbohydrases are largely used in feed formulation with an energy matrix. Their individual biological effects are well defined, and these additives act differently: one saves energy that would be spent in the immune response (β-mannanases), and the others release nutritional components for absorption (arabinofuranosidases, β-glucanase, and xylanases). Due to these complementary effects, we hypothesized that the supplementation of β-mannanase, arabinofuranosidases, β-glucanase, and xylanases in nutritionally balanced diets would be complementary and their combination would produce a more pronounced effect on the metabolism and health of pigs. This information can assist nutritionists in formulating diets in the practical production context while contributing to more efficient animal production systems. Therefore, our objective was to evaluate whether the addition of β-mannanase alone or combined with the multi-carbohydrase complex could act additively to improve diet digestibility, nutrient, and energy metabolism, as well as intestinal health in growing pigs.

## 2. Material and Methods

### 2.1. Enzymes

Both the enzymes used in the study were commercially available and were directly acquired from a feed meal company, ensuring that the results closely reflected field scenarios. The β-mannanase tested was derived from the fermentation of *Paenibacillus lentus* (Hemicell HT, Elanco Animal Health, São Paulo, Brazil; Brazilian registration number: SP626-2; minimum activity: 160,000 unit/g (1 unit is the amount of enzyme that releases 0.72 mcg of reducing sugars (equivalent to D-mannose) per min from goma locust (mannan concentration of 88%) at pH 7.5 and 40 °C)). The estimated concentration of β-mannans in the experimental feeds was 0.528% [14].

The multi-carbohydrase complex tested was produced from *Talaromyces versatilis* (Rovabio Advance, Adisseo, São Paulo, Brazil; Brazilian registration number: SP3557-2; minimum activity: 1250 visco-units of endo-β-1,4-xylanase, 860 visco-units of endo-1,3(4)-β-glucanase (1 visco-unit of xylanase or β-glucanase activity is defined as the amount of enzyme that is hydrolyzed by the substrate (wheat AX and barley β-glucan, respectively) reducing the viscosity of the solution, resulting in a change in relative fluidity of 1 arbitrary unit per min per mL at pH 5.5 and 30 °C), and 9250 visco-units of α-L-arabinofuranosidase (1 visco-unit refers to the amount of enzyme that releases 1 nmol of arabinose per min from the hydrolysis of wheat AX at pH 4 and 50 °C).

### 2.2. Animals, Diets, and Experimental Procedures

The experiment was carried out in the experimental facilities at the Universidade Federal do Rio Grande of Sul, located in Porto Alegre, Rio Grande of Sul, Brazil. The number of replications was calculated using expected variability (obtained from previous studies developed in the laboratory) in the metabolizable energy values [15]. Thus, a total of 24 barrows (Large White × Landrace; 8 pigs per treatment) with an average initial body weight (BW) of 35.56 kg (±3.81 kg) were individually housed in metabolism crates (1.66 m long × 0.63 m wide × 1.58 m high).

The average room ambient temperature was 17.1 °C and the daily relative humidity averaged 70%. These values suggested that the pigs were housed under thermoneutral conditions. No pigs were removed from the trial and no health issues were detected during the experimental period.

The pigs were randomly assigned to one of three enzyme supplementation plans: control (no addition), the addition of β-mannanase (300 g/ton; BM), or β-mannanase plus multi-carbohydrase complex, including xylanase, β-glucanase, and arabinofuranosidases (300 g/ton + 50 g/ton; BM + MCC). The pigs remained in the experiment for 13 days, which consisted of a 5-day adaptation period and a subsequent 8-day collection period. No pigs were removed from the trial and no health issues were detected during the experimental period.

The experimental feeds were formulated for the minimum cost solution to meet the nutritional requirements recommended by the Brazilian Tables for Poultry and Swine (Table 1; [2]). The same reference was used for ingredient composition, except for soybean meal and corn, for which the total energy and crude protein content were analyzed and later used to estimate the metabolizable energy and digestible amino acid levels [2].

Three modifications were performed in the control formula to obtain the addition treatments. First, β-mannanase or β-mannanase plus the multi-carbohydrase complex was included in the formulas depending on the treatment. In addition, the inclusion of soybean oil in the addition diets was adjusted to reduce the metabolizable energy level by 90 kcal, which is comparable to the energy matrix commonly attributed to enzymes in practical conditions. Later, a final adjustment was performed using an inert material (i.e., washed sand was added until the sum of the ingredient levels reached 100%) to complete the formula.

Phytase was included (with a matrix for Ca and P of 500 FTU/kg) in all the treatments to better represent standard commercial feeding programs. The analyzed composition of the feed samples was validated before the trial.

### 2.3. Data Collection

#### 2.3.1. Digestibility and Metabolism

The metabolism crates were equipped with trays for the total collection of feces and a system for the total collection of urine. Water and mash form feed were provided ad libitum throughout the adaptation periods. During the collection period, the pigs received feed according to their metabolic body weight (2.6 × the estimated maintenance requirement; [16]).

During the collection period, the diet provided was quantified and the feed samples were collected, identified, and stored in a freezer for further analysis. The feces and urine samples were collected twice daily (8:00 a.m. and 5:30 p.m.). The beginning and end of the collection period were defined using an indigestible marker (1.5% ferric oxide) mixed in the diets. All the samples were stored in plastic bags identified by the experimental unit and stored in a freezer (−20 °C).

At the end of the experimental period (day 13), the fecal and urine samples were thawed at room temperature, weighed, and homogenized. The samples from each experimental unit were collected and lyophilized. Then, the samples of feed, feces, and urine were analyzed for dry matter (oven at 105 °C), nitrogen (micro Kjeldahl method), and gross energy (calorimetric pump) following the procedures described by AOAC [17]. The coefficients of digestibility (dry matter, protein, and energy) and metabolizability (protein and energy), in addition to the apparent metabolizable energy values, were calculated from the data obtained according to the equations provided by Sakomura and Rostagno [15]. Manure production was also estimated as the total volume of urine and feces collected during the study period. Figure 1 describes the timeline of all sampling in growing pigs fed different enzyme additions.

#### 2.3.2. Feed Retention Rate and Fecal Moisture Content

The time spent from the consumption of feed with ferric oxide to the appearance of the first marked feces was recorded at the beginning and end of each trial following the procedure described by Moore and Winter [18]. The feces collected each day were weighed and homogenized. A sample corresponding to 10% of the total weight was separated, and the dry matter content was analyzed in oven at 105 °C.

#### 2.3.3. Acute-Phase Proteins

On day 13, all the pigs were fasted for 8 h and a volume of 5 mL of blood was collected from the vena cava into sterile vacutainer tubes without anticoagulant. The samples were identified and held in a thermal box with ice for 40 min before centrifugation. The samples were centrifuged at 3500 rpm for 10 min and the serum was separated, collected, and frozen (−20 °C) for acute-phase protein analysis. Serum haptoglobin, transferrin, and C-reactive protein concentrations were determined by sodium dodecyl sulfate-polyacrylamide gel electrophoresis [19]. The molecular weight and protein fraction concentrations were determined using computer densitometry (Shimadzu 9301 PC, Shimadzu Corp, Kyoto, Japan) with a basic scanner. The proteins were identified using biomarkers (Sigma Marker, Sigma-Aldrich Biotechnology LP, Steinheim, Germany). For the densitometric examination of the protein bands, reference curves were created from the reading of the standard marker. Afterward, the concentrations were normalized to the total protein serum concentration determined using commercial reagents (Wiener lab, São Paulo, Brazil).

#### 2.3.4. Fecal Biomarkers

After homogenizing all the stool samples collected during the digestibility period, another sample was taken for biomarker assessment. Briefly, 5 mL of phosphate-buffered saline (PBS) was added to each tube containing a subsample of 100 mg of feces and mixed thoroughly. After that, the tubes were centrifuged for 25 min at 400 rpm. One mL of each sample was transferred to a microtube and centrifuged for 20 min at 1500 rpm. The supernatant was then collected and stored at −20 °C until analysis. The detection of porcine calprotectin, neopterin, lactoferrin, calcium-binding proteins, and intestinal fatty acid binding protein was performed using commercial ELISA tests (My Biosource, San Diego, CA, USA) following the manufacturer’s protocol (Multiskan Sky, Thermo Scientific, São Paulo, Brazil).

#### 2.3.5. Intestinal Morphometry and Intestinal Rupture Resistance

All the pigs were slaughtered on day 13 of the experiment following the animal welfare and euthanasia standards outlined in the CONCEA euthanasia practice guidelines [20] using electrical stunning methods. Intestinal samples (4 cm distal to the stomach for the duodenum, mid jejunum, and 4 cm distal to the jejunum for the ileum) were collected and stored in vials with 10% formaldehyde solution. Slides with histological sections were prepared and stained with hematoxylin and eosin. In the intestinal fragments, the height of the villi, the perimeter, and the depth of the intestinal crypts were evaluated according to the methodology described by Galli et al. [21].

Other segments of the jejunum and colon (four samples per pig, approximately 5 cm in length per segment) were randomly collected immediately after slaughter. These samples were then used to assess intestinal rupture strength using a dynamometer (ITFG6005; Instrutemp, São Paulo, Brazil) that provides the ideal force necessary to break the sample [22]. The results were expressed as kilogram-force per centimeter square (kgf/cm).

### 2.4. Statistical Analysis

All data were submitted to the Ryan–Joiner test to assess their normal distribution. The analysis of variance was performed using the General Linear Model using the Minitab 21 software v2021 (State College, PA, USA). The statistical model considered the fixed effect of treatment and the error. Body weight was considered as a co-variable. Fecal moisture was also analyzed by considering the effect of sampling time (days) in the model. Eventual differences among the treatments were assessed with the Tukey multiple comparison test and then interpreted at *p* ≤ 0.05 (significant differences) and 0.05 < *p* ≤ 0.10 (a trend for difference).

## 3. Results

### 3.1. Digestibility and Metabolism

No differences were found among the treatments for feed allotment, leftovers, and intake (Table 2). However, BM and BM + MCC resulted in greater dry matter coefficients of digestibility compared to the control (*p* < 0.01; Table 3). BM and BM + MCC also improved the digestibility and metabolizability coefficients of energy and protein (*p* < 0.01) compared to the control. No differences between BM alone or combined (BM + MCC) were found for these responses.

No differences (*p* > 0.10) among the enzyme addition were found for the energy balance responses, despite a reduction in the fecal energy content in the pigs with BM and BM + MCC when compared to the control (*p* < 0.01; Table 4). To interpret these results, it is important to consider that an energy matrix was applied when formulating the addition treatments.

No differences were found among the enzyme addition for intake, fecal, urinary, and absorbed nitrogen (*p* > 0.10; Table 5). However, the BM treatment tended to improve nitrogen retention relative to control (*p* = 0.09). In addition, a greater ratio between retained and absorbed nitrogen was observed in the treatments with BM and BM + MCC compared to the control (*p* < 0.01).

### 3.2. Production of Manure, Feed Retention Rate, and Fecal Moisture Content

Fecal and urinary volumes did not differ (*p* > 0.10) among the treatments when assessed independently (Table 6). However, when assessed combined, lower manure production was observed in both the enzyme additions compared to the control (*p* < 0.01).

No difference (*p* > 0.10) was found among the treatments for the feed retention rate. Diarrhea was not observed during this trial. However, the fecal moisture was greater in the control (mean value: 69.42%) than in the BM + MCC (65.63%) and BM (65.63%; Figure 2) treatments.

### 3.3. Acute-Phase Proteins and Fecal Biomarkers

No differences (*p* > 0.10) among the treatments were found for the serum acute-phase proteins C-reactive protein, transferrin, and haptoglobin, as well as for the fecal biomarkers neopterin, lactoferrin, calcium-binding proteins, and intestinal fatty acid binding protein (Table 7). However, the use of BM and BM + MCC resulted in remarkably lower fecal calprotectin concentrations compared to the control (52 and 56% reduction, respectively; *p* = 0.099). C-reactive protein (average for all the treatments 0.800 mg/dL) and transferrin (average for all the treatments 35 mg/dL) were removed from the table because the measure did not work.

### 3.4. Intestinal Rupture Resistance and Morphology

The BM and BM + MCC use resulted in a tendency for greater intestinal rupture resistance of the jejunum compared to the unadded group (*p* = 0.08; Table 8). In the colon, the resistance was greater in the BM + MCC pigs compared with the BM and control (*p* < 0.05), and greater in the BM than in the control pigs (*p* < 0.05).

The effects of each enzyme addition on morphological assessments were not constant among the intestinal regions. A greater villus height in the duodenum was observed in the BM and BM + MCC treatments than in the control treatment (*p* < 0.05). The BM + MCC addition also resulted in a lower villus diameter compared to the BM and control (*p* < 0.01). A greater crypt depth was observed in the BM and BM + MCC in relation to the control (*p* < 0.01). No difference was found among the treatments for villi area and villi height–crypt depth ratio.

In the jejunum, a greater villus height was observed in the BM treatment, followed by the BM + MCC, and then the control treatment (*p* < 0.01). The BM use resulted in a greater villus area and villi height to crypt depth ratio compared to the control (*p* < 0.05). No differences were found among the treatments for villi width and crypt depth (*p* > 0.10).

Finally, in the ileum, a greater villus height was observed in the BM addition than in the control group (*p* < 0.01). A tendency for a greater villus area was also observed in the BM than in the other treatments (*p* = 0.097). No difference was found among the treatments for villi width, crypt depth, and the ratio of villi height to crypt depth (*p* > 0.10).

## 4. Discussion

This study was conducted to evaluate whether the addition of BM alone or combined with the multi-carbohydrase complex (BM + MCC) can act additively to improve diet digestibility, nutrient metabolism, and gut health in growing pigs. The hypothesis tested in this study is that even if both the enzymes are used with an energy matrix in feed formulation, these additives have different action mechanisms and could have complementary effects on pigs’ metabolism. Β-mannanase is responsible for saving energy that would be spent triggering an immune response, whereas other carbohydrates, such as arabinofuranosidases, β-glucanase, and xylanases, release trapped nutrients from plant cells to be absorbed. To the best of our knowledge, there are no other studies with a similar design for these enzyme combinations. A large amount of research effort has been made to determine nutritional strategies to improve gut health in post-weaned pigs compared to growing pigs, especially regarding fecal biomarkers.

In the current study, the addition of both enzymes (BM or BM + MCC) resulted in the greater digestibility of dry matter, protein, and crude and metabolizable energy. The improvement in the nutrient digestibility coefficient may be related to the reduction in molecule size with enzyme addition, which allows for a wider contact area for nutrient absorption [23]. In agreement, Pettey et al. [24] observed that the dietary addition of 0.05% BM for pigs saved the equivalent of 100 kcal/kg of metabolizable energy, which may be combined with an increased efficiency in energy use. Furthermore, the addition of 200 mL/ton of an enzyme blend (xylanases and arabinofuranosidases) improved the total tract digestibility of organic matter, energy, protein, crude fiber, and starch in growing pigs [25]. The authors related the improved nutrient digestibility to the increased access of endogenous proteolytic, amylolytic, and lipolytic enzymes to nutrients. Therefore, the greater digestibility observed in this study may be attributed to the enzymes having better access to the substrates, allowing for a greater amount of nutrients to be absorbed. Indeed, the xylanase enzyme can release encapsulated nutrients in plants and modulate the microbiota through the prebiotic action of the oligosaccharides liberated from arabinoxylan hydrolysis [26].

Greater nitrogen retention and an improved ratio of retained to absorbed nitrogen were observed in the pigs that received both the enzymes. Similar results were reported by Hlongwana et al. [27], who observed greater efficiency in the use of nitrogen ingested by pigs fed with Amarula cake compared with the control treatment. These results can be attributed to the pigs’ greater muscle growth, as well as the synthesis of intestinal cells.

Furthermore, the pigs that received the BM and BM + MCC had lower fecal energy than the control, which may be explained by the improved digestibility of crude and metabolizable energy. In addition, β-mannanase- and xylanase-enhanced NSP hydrolysis into smaller units results in additional energy for pig metabolism [28,29]. Excess energy can increase manure production [30] such as observed in the control treatment.

In agreement with the previous paragraph, Sánchez-Uribe et al. [31] reported that the dietary addition of 0.03% BM for growing and finishing pigs can reduce 1.6% net energy compared to the control. Genova et al. [32] observed that 0.03% BM plus xylanase-phytase for finisher pigs saved 85 kcal of ME/kg out of improved gain–feed ratio, energy, and protein usage without metabolic and intestinal ecosystem disorders.

β-mannanase alone or combined with a multi-carbohydrase complex resulted in lower manure production compared to the control treatment. To our knowledge, there are still no studies available in the literature showing this effect with the use of β-mannanase alone or combined with a multi-carbohydrase complex. Lower manure production is important in the context of more sustainable animal production. Furthermore, xylanases and β-mannanase dietary addition have been shown to lower intestinal viscosity produced by high dietary NSP concentrations [33,34]. Therefore, it reduces the viscosity of the digesta, increases the availability of nutrients, and minimizes the presence of undesirable fermentable substances in the distal small intestine [9,10,11,12,13,14,15,16,17,18,19,20,21,22,23,24,25,26,27,28,29,30,31,32,33,34,35]. Collectively, these enzymes can have a greater nutrient absorption by enterocytes [36].

We found that BM and BM + MCC had lower fecal moisture content compared to the control. In this context, Sánchez-Uribe et al. [31] observed that the dietary addition of 0.03% BM to growing and finishing pigs reduced diarrhea and local immune stimulation in relation to the control treatment (without enzyme). In this case, the NSP skeleton has a water-retaining capacity that forms a gelatinous structure around the feed bolus and increases the viscosity of the digest [36,37,38]. Therefore, the results of this study can be explained by the fact that the enzymes reduced the NSP content in the digesta, which left fewer fibers accessible in the digestive tract to bind to water and form a gelatinous structure, which may result in the lowest moisture in the feces.

Intestinal morphology is one of the parameters that is related to gut health [39]. In the current study, the addition of BM alone improved the intestinal morphology of the duodenum, jejunum, and ileum in growing pigs. Jang et al. [40] reported that dietary supplementation with 0.06% β-mannanase for post-weaned pigs resulted in greater intestinal villi and villi–crypt ratio, and lower crypt depth of the jejunum in relation to non-enzyme addition pigs, which may be attributed to lower NSP content and digesta viscosity. Also, Jang et al. [41] observed that 600 U/kg β-mannanase in a corn–soybean-based diet improved intestinal morphology via villus height, the ratio of villus height to crypt depth, and crypt cell proliferation in the jejunum and duodenum, as well as the enzyme hydrolysis molecules into a smaller degree of polymerization and that effect explains the decreased digesta viscosity. Therefore, the hydrolysis of β-mannans explains better nutrient and energy digestibility, and intestinal health. However, we observed that BM combined with a multi-carbohydrase complex showed results that were more complex to interpret, with lower jejunal villus height and ileal villus area compared with BM alone. Our findings do not corroborate those of Moita et al. [42], who reported that increasing xylanase (0, 220, 440, 880, and 1760 xylanase units per kg feed) improved intestinal morphology via villus height and reduced the viscosity of jejunal digesta. Although both the enzymes can improve gut morphometry when used alone, their combined effects can lead to changes in substrate availability, microbial populations, mucus layer, and/or gut motility, all of which can potentially affect gut morphology [43]. Unfortunately, studies comparing the effects of the same enzyme when provided alone versus in combination with other enzymes or additives are lacking [44]. In addition, the available results are frequently controversial, as the effects can vary based on the characteristics of the animal (e.g., age) and diet (e.g., substrate availability). Therefore, the careful monitoring of the effects of combined enzymes in future research is crucial to mitigate potential negative impacts on gut health.

The intestinal epithelium serves as a physical barrier between the pathogens and the gastrointestinal tract. These epithelial cells are connected by a complex of junctions that consist of tight junctions, adherents’ junctions, gap junctions, and desmosomes [45]. In this study, the high intestinal resistance of the jejunum and colon of the pigs that received both enzyme addition might be related to an improvement in this line of defense resulting in lower intestinal permeability.

Immune system stimulation causes the release of the calcium–zinc-binding protein calprotectin, which belongs to the S100 family [46]. Neutrophils are immune system cells that release calprotectin, which indicates that it can be used as a non-invasive marker to verify intestinal inflammation [47]. Chang et al. [48] demonstrated that calprotectin downregulation may be a sign of reduced intestinal inflammation in *E. coli*-infected post-weaned pigs. In the present study, remarkably lower calprotectin concentrations were observed with the enzyme treatments (BM and BM + MCC), which may be associated with a lower inflammation process caused by anti-nutritional factors. The literature on this fecal biomarker in growing pigs is still lacking data; thus, more research is needed on pigs’ challenges (feed, pathogens, and weather). The great advantage of BM addition is to reduce the immune system over-stimulation, avoiding nutrients that were shifted to generate immune response molecules rather than growth [49].

Finally, the BM alone or combined with the multi-carbohydrase complex showed positive effects on the feed digestibility, metabolism of nutrients, and gut health in the growing pigs. Additional effects of these enzymes were not found for most of the responses assessed in this study, probably because the diets did not have enough substrates (xylans, glucans, and/or arabinoxylans) to result in an additive effect especially since these NPS are found in the corn. This may be explained by the lower inclusion or lower amount of NPS in corn in the diets. Therefore, despite our initial hypothesis that an additional effect could be observed, it is possible that the use of other ingredients with higher NSP content may result in additive enzymatic effects. Galli et al. [50], when assessing the same enzyme plans in simple and complex diets for nursery pigs, observed complementary effects on digestibility and intestinal health, indicating the need for more studies assessing the complementarity of these enzymes in other feeding contexts (i.e., with other feed ingredients). Collectively, the effects of a combination of enzymes depend on several factors such as enzyme specificity towards the target substrate, dosage levels, interactions between different enzymes, quality, and composition of ingredients, as well as the age of the animals [51].

Research projects that investigate the effects of enzymes when used alone or in combination with other enzymes (or even other feed additives) are of paramount importance. In the context of the current study, the inclusion of an additional treatment to evaluate the effect of the multi-carbohydrase complex, when supplied alone, would enhance our understanding of enzyme interactions during statistical analysis. In addition, exploring the same enzyme combinations in feeds formulated with different ingredient matrices, such as those containing wheat or barley, presents an intriguing avenue for investigation. However, both of these endeavors would necessitate an increase in the number of animals and other resources, including personnel, which were not available during the development of this study (due to the COVID-19 pandemic). Future research should address these limitations to contribute to the understanding of enzyme efficacy in various feed formulations.

Additionally, the effects of using enzymes individually or in combination need to be further evaluated in long-term experiments. The conditions required in traditional digestibility trials (e.g., individual housing in metabolic crates) cannot be maintained for extended periods of time. Therefore, the duration of the trial is not a limitation of this particular study; rather, it is inherent in the traditional methodology used for determining digestibility and metabolism coefficients in pigs. Nevertheless, longer experiments are crucial for assessing other relevant responses such as growth performance and gut microbiota stability.

Modern pig production is moving towards more precise and sustainable practices. The use of combined enzymes in feed formulations is certainly a part of this context. Thus, although most of the matrices available in the literature were generated in studies developed with a single addition, this is not representative of most commercial pig diets in which several enzymes are used combined. Enzymes are certainly one of the most studied topics in pig nutrition. Still, more studies testing their combinations are crucial to allow even more precise use in feed formulation.

## 5. Conclusions

β-mannanase alone or combined with the multi-carbohydrase complex (arabinofuranosidases, β-glucanase, and xylanases) can be used as a nutritional strategy to improve the digestibility of nutrients, energy metabolism, and gut health which contributes to reducing nitrogen emissions and manure production from swine production. Therefore, testing alternative ingredients (sorghum and wheat) and these enzyme combinations, as well as growth performance, can be beneficial for future studies.

## Figures and Tables

**Figure 1 animals-14-03457-f001:**
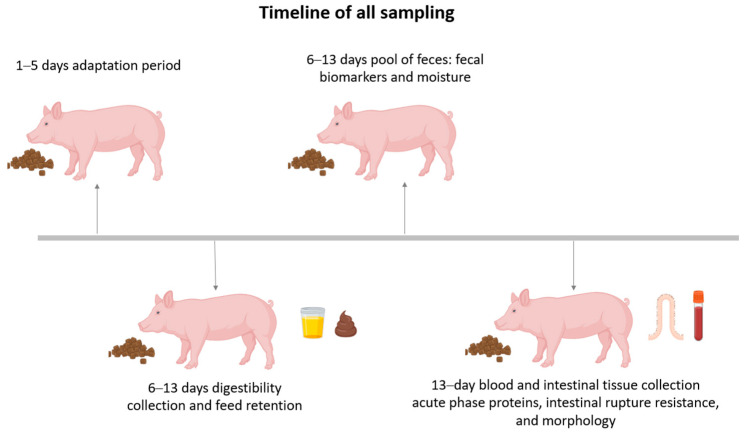
Timeline of all sampling in growing pigs fed different enzyme additions.

**Figure 2 animals-14-03457-f002:**
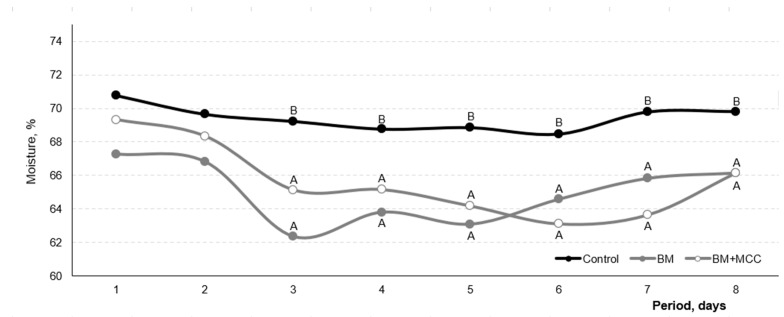
Fecal moisture content observed in the growing pigs with enzyme addition. Enzyme addition: Without any carbohydrase (control) or the addition of β-mannanase (BM; 300 g/ton) or β-mannanase plus the multi-carbohydrase complex (BM + MCC; 300 g/ton + 50 g/ton) (8 pigs per treatment). Period days 1–8 represent the digestibility period. *p*-value: Probabilities for the effects of enzyme addition. Means followed by different uppercase letters differ statistically at *p* ≤ 0.05 level.

**Table 1 animals-14-03457-t001:** Ingredient formulas and chemical composition of experimental feeds used in this trial, presented according to the enzyme addition ^1^.

	Enzyme Addition		
Enzyme Addition	Control	BM	BM + MCC
**Ingredients, %**			
Corn	58.61	58.61	58.61
Soybean meal 46%	35.45	35.45	35.45
Soybean oil	2.916	1.832	1.832
Salt	0.442	0.442	0.442
Limestone	0.910	0.910	0.910
Phosphate	0.787	0.787	0.787
Premix ^2^	0.500	0.500	0.500
L-Lysine	0.202	0.202	0.202
DL-Methionine	0.102	0.102	0.102
L-Threonine	0.066	0.066	0.066
Phytase ^3^	0.005	0.005	0.005
Inert (washed sand)	-	1.054	1.049
Beta-mananase ^4^	-	0.030	0.030
Multi-enzyme blend ^5^	-	-	0.005
**Calculated composition**			
Crude protein, %	20.94		
Digestible lysine %	1.157		
Metab. energy, kcal/kg	3350	3260	3260
Total calcium, %	0.769		
Total phosphorus, %	0.493		
Available phosphorus, %	0.380		

^1^ enzyme addition: Without any carbohydrase supplementation (control) or the addition of β-mannanase (BM; 300 g/ton) or β-mannanase plus the multi-carbohydrase complex (BM + MCC; 300 g/ton+50 g/ton). ^2^ premix with vitamins and minerals per kg/feed. Vitamin A: 6.400 UI/kg; vitamin B1: 0.900 mg/kg; vitamin B2: 0.0057 g/kg; vitamin K3 3.300 mg/kg; vitamin B6: 1.800 mg/kg; vitamin B12 26.00 mcg/kg; vitamin D3: 1.325 UI/kg; vitamin E: 36.10 UI/kg; folic acid 0.300 mg/kg; niacin: 0.040 g/kg; biotin 0.100 mg/kg; pantothenic acid: 0.020 g/kg; calcium 0.045 g/kg; phosphorus 0.010 g/kg; sodium 0.017 g/kg; copper 0.500 mg/kg; iron 0.500 mg/kg; iodine 0.01 mg/kg; manganese 0.250 mg; zinc 0.007 g/kg. ^3^ Natuphos (BASF Corporation, São Paulo, Brazil; minimum activity of 500 FTU per kg of feed). ^4^ Hemicell HT (Elanco Animal Health, São Paulo, Brazil; minimum activity of 48,000 U of β-mannanase per kg of feed). ^5^ Rovabio Advance (Adisseo, São Paulo, Brazil; minimum activity of 62.50 visco-U of xylanase, 43.00 visco-U of β-glucanase, and 462.5 visco-U of arabinofuranosidase per kg of feed).

**Table 2 animals-14-03457-t002:** Feeding responses of growing pigs fed different enzyme addition ^1^.

Variable	Enzyme Addition			RSE ^2^	*p*-Value ^3^
	Control	BM	BM + MCC		
Feed allotment (g/day)	1666	1766	1693	106.0	0.131
Intake (g/day)	1606	1682	1576	0.970	0.125
Leftovers (g/day)	59.10	60.38	114.5	74.99	0.252

^1^ enzyme addition: Without any carbohydrase (control) or the addition of β-mannanase (BM; 300 g/ton) or β-mannanase plus the multi-carbohydrase complex (BM + MCC; 300 g/ton + 50 g/ton) (8 pigs per treatment). ^2^ RSE: Residual Standard Error. ^3^ *p*-value: Probabilities for the effects of enzyme addition. Differ statistically at *p* ≤ 0.05 level, while a trend at 0.05 < *p* ≤ 0.10.

**Table 3 animals-14-03457-t003:** Coefficients of digestibility and metabolizability observed in growing pigs with enzyme addition ^1^.

Variable	Enzyme Addition			RSE ^2^	*p*-Value ^3^
	Control	BM	BM + MCC		
Dry matter digestibility (%)	90.57 ^B^	92.39 ^A^	91.56 ^A^	0.898	0.002
Protein digestibility (%)	89.84 ^B^	92.42 ^A^	91.87 ^A^	1.232	0.001
Energy digestibility (%)	90.03 ^B^	92.68 ^A^	91.71 ^A^	0.874	<0.001
Protein metabolizability (%)	89.42 ^B^	91.76 ^A^	90.94 ^A^	1.336	0.005
Energy metabolizability (%)	87.90 ^B^	90.19 ^A^	89.17 ^A^	0.972	<0.001

^1^ enzyme addition: Without any carbohydrase (control) or the addition of β-mannanase (BM; 300 g/ton) or β-mannanase plus the multi-carbohydrase complex (BM + MCC; 300 g/ton + 50 g/ton) (8 pigs per treatment). ^2^ RSE: Residual Standard Error. ^3^ *p*-value: Probabilities for the effects of enzyme addition. Means followed by different uppercase letters differ statistically at *p* ≤ 0.05 level.

**Table 4 animals-14-03457-t004:** Energy balance observed in growing pigs with enzyme addition ^1^.

Variable	Enzyme Addition			RSE ^2^	*p*-Value ^3^
	Control	BM	BM + MCC		
Intake (kcal/day)	5852	5974	5598	364.6	0.139
Fecal (kcal /day)	579.1 ^A^	504.5 ^B^	515.7 ^B^	44.11	0.003
Urinary (kcal /day)	122.1	147.1	152.7	40.87	0.254
DE ^4^ (kcal/kg)	3280	3287	3269	29.78	0.544
ME ^5^ (kcal/kg)	3202	3203	3187	38.72	0.654
Ratio EM/ED (%)	97.87	98.39	98.29	0.506	0.116

^1^ enzyme addition: Without any carbohydrase (control) or the addition of β-mannanase (BM; 300 g/ton) or β-mannanase plus the multi-carbohydrase complex (BM + MCC; 300 g/ton+50 g/ton) (8 pigs per treatment). ^2^ RSE: Residual Standard Error. ^3^ *p*-value: Probabilities for the effects of enzyme addition. Means followed by different uppercase letters differ statistically at *p* ≤ 0.05 level. ^4^ DE: digestible energy. ^5^ ME: metabolizable energy.

**Table 5 animals-14-03457-t005:** Nitrogen balance observed in growing pigs with enzyme addition ^1^.

Variable	Enzyme Addition			RSE ^2^	*p*-Value ^3^
	Control	BM	BM + MCC		
Intake (g/day)	49.50	51.82	48.56	3.110	0.125
Fecal (g/day)	4.672	4.521	4.453	0.489	0.658
Urinary (g/day)	0.199	0.202	0.280	0.158	0.582
Absorbed (g/day)	44.51	47.94	45.80	3.521	0.112
Retained (g/day)	44.32 ^b^	47.61 ^a^	45.11^ab^	3.325	0.092
Ratio ret./abs. ^4^ (%)	99.52 ^B^	100.3 ^A^	100.2 ^A^	0.524	0.005

^1^ enzyme addition: Without any carbohydrase (control) or the addition of β-mannanase (BM; 300 g/ton) or β-mannanase plus the multi-carbohydrase complex (BM + MCC; 300 g/ton + 50 g/ton) (8 pigs per treatment). ^2^ RSE: Residual Standard Error. ^3^ *p*-value: Probabilities for the effects of enzyme addition. Means followed by different uppercase letters differ statistically at *p* ≤ 0.05 level, while lowercase letters are used to indicate a trend at 0.05 < *p* ≤ 0.10. ^4^ Ratio ret./abs.: the ratio between protein retention and protein absorption.

**Table 6 animals-14-03457-t006:** Manure production (dry matter basis) and feed retention rate observed in growing pigs with enzyme addition ^1^.

Variable	Enzyme Addition			RSE ^2^	*p*-Value ^3^
	Control	BM	BM + MCC		
Feces (g/day)	134.4	129.7	132.3	11.90	0.698
Urine (g/day)	60.42	75.12	78.60	18.61	0.107
Manure (g/day)	6728 ^A^	4724 ^B^	3379 ^B^	1462	0.001
Feed retention rate (min)	1998	1600	1600	757.5	0.365

^1^ enzyme addition: Without any carbohydrase (control) or the addition of β-mannanase (BM; 300 g/ton) or β-mannanase plus the multi-carbohydrase complex (BM + MCC; 300 g/ton + 50 g/ton) (8 pigs per treatment). ^2^ RSE: Residual Standard Error. ^3^ *p*-value: Probabilities for the effects of enzyme addition. Means followed by different uppercase letters differ statistically at *p* ≤ 0.05 level.

**Table 7 animals-14-03457-t007:** Serum acute-phase proteins and fecal biomarkers observed in growing pigs with enzyme addition ^1^.

Variable	Enzyme Addition			RSE ^2^	*p*-Value ^3^
	Control	BM	BM + MCC		
C-reactive protein (mg/dL)	0.800	0.800	0.800	-	-
Transferrin (mg/dL)	35.00	35.00	35.00	-	-
Haptoglobin (mg/dL)	9.600	10.80	9.000	2.100	0.448
Calprotectin (ng/mL)	37.33 ^a^	18.00 ^b^	16.56 ^b^	14.97	0.099
Neopterin (ng/mL)	0.188	0.182	0.172	0.032	0.789
Lactoferrin (ng/mL)	0.392	0.392	0.392	0.000	0.704
Calcium-binding proteins (pg/mL)	108.5	85.71	82.14	27.11	0.305
Intestinal fatty acid binding protein (pg/mL)	42.12	43.13	43.19	2.916	0.819

^1^ enzyme addition: Without any carbohydrase (control) or the addition of β-mannanase (BM; 300 g/ton) or β-mannanase plus the multi-carbohydrase complex (BM + MCC; 300 g/ton + 50 g/ton) (8 pigs per treatment). ^2^ RSE: Residual Standard Error. ^3^ *p*-value: Probabilities for the effects of enzyme addition. Means followed by different lowercase letters are used to indicate a trend at 0.05 < *p* ≤ 0.10.

**Table 8 animals-14-03457-t008:** Intestinal rupture resistance and morphometry observed in growing pigs with enzyme addition ^1^.

Variable	Enzyme Addition			RSE ^2^	*p*-Value ^3^
	Control	BM	BM + MCC		
**Intestinal rupture resistance**					
Jejunum (kgf)	3.430 ^b^	3.701 ^a^	3.601 ^a^	0.178	0.083
Colon (kgf)	2.855 ^C^	3. 144 ^B^	3.531 ^A^	0.460	0.042
**Duodenal morphology**					
Villi height (μm)	510.8 ^B^	533.5 ^A^	577.1 ^A^	14.20	0.030
Villi width (μm)	143.4 ^A^	138.9 ^A^	121.2 ^B^	2.540	<0.001
Villi area (μm^2^)	7426	7423	7013	4998	0.765
Crypt depth (μm)	419.3 ^B^	497.9 ^A^	471.0^A^	18.72	0.004
Ratio villi height to crypt depth	1.224	1.090	1.307	0.082	0.111
**Jejunal morphology**					
Villi height (μm)	430.8 ^C^	549.9 ^A^	494.3 ^B^	30.96	<0.001
Villi width (μm)	122.0	124.8	129.0	0.468	3.629
Villi area (μm^2^)	5347 ^B^	6983 ^A^	6231 ^AB^	4374	0.003
Crypt depth (μm)	292.6	296.4	319.3	11.25	0.261
Ratio villi height to crypt depth	1.558 ^B^	2.013 ^A^	1.704 ^AB^	0.154	0.012
**Ileal morphology**					
Villi height (μm)	388.9 ^B^	483.2 ^A^	395.5 ^AB^	20.62	0.002
Villi width (μm)	120.500	125.630	122.610	23.82	0.964
Villi area (μm^2^)	46857 ^b^	59954 ^a^	46137 ^b^	3065	0.097
Crypt depth (μm)	252.6	277.7	218.8	177.3	0.128
Ratio of villi height to crypt depth	1.573	3.011	2.950	1.932	0.874

^1^ enzyme addition: Without any carbohydrase (control) or the addition of β-mannanase (BM; 300 g/ton) or β-mannanase plus the multi-carbohydrase complex (BM + MCC; 300 g/ton + 50 g/ton) (8 pigs per treatment). ^2^ RSE: Residual Standard Error. ^3^ *p*-value: Probabilities for the effects of enzyme addition. Means followed by different uppercase letters differ statistically at *p* ≤ 0.05 level, while lowercase letters are used to indicate a trend at 0.05 < *p* ≤ 0.10.

## Data Availability

The original contributions presented in the study are included in the article, further inquiries can be directed to the corresponding authors.

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
