# Peer review of "Effects of β-Mannanase Alone or Combined with Multi-Carbohydrase Complex in Corn–Soybean Meal Diets on Nutrient Metabolism and Gut Health of Growing Pigs"

_animals, 2024, doi:10.3390/ani14233457_

Round 1
Reviewer 1 Report (Previous Reviewer 1)
Comments and Suggestions for Authors
The manuscript has been well improved, and I think most of my comments have been addressed. The manuscript now demonstrates much more clarity; the title has greater transparency, and the objectives are well presented. The additional information regarding the Introduction, M&M and Discussion sections provides a well-balanced and comprehensive article. Generally, accuracy and attention to detail have been greatly improved.
Author Response
Dear Editor and Reviewers,
We do appreciate your consideration in reviewing the manuscript. Thank you for taking the time to evaluate it. All the suggestions have been carefully considered in the revised version of the manuscript. Our answers are listed below and an updated version of the manuscript with all the modifications (i.e., corrections and additional information inserted) highlighted in red is provided.
The surname of the author Alicia Zem Fraga was typed wrong. The correct is "Zem" instead of “Zam”.
Yours sincerely,
Gabriela Miotto Galli, main author
Ines Andretta, correspondent author
EDITOR
EDITOR: Table 1: In the footnote #2, the vitamin and trace element concentrations need to be provided as per kilogram of mixed diet. The present values appear to be to concentrations in the premix.
ANSWER: Yes, thank you so much for point out this error.
We added: 2 Premix with vitamins and minerals per kg/feed. Vitamin A: 6.400 UI/kg; vitamin B1: 0.900 mg/kg; vitamin B2: 0.0057 g/kg; vitamin K3 3.300 mg/kg; vitamin B6: 1.800 mg/kg; vitamin B12 26.00 mcg/kg; vitamin D3: 1.325 UI/kg; vitamin E: 36.10 UI/kg; folic acid 0.300 mg/kg; niacin: 0.040 g/kg; biotin 0.100 mg/kg; pantothenic acid: 0.020 g/kg. calcium 0.045 g/kg; phosphorus 0.010 g/kg; sodium 0.017 g/kg; copper 0.500 mg/kg; iron 0.500 mg/kg; iodine 0.01 mg/kg; manganese 0.250 mg/kg; zinc 0.007 g/kg.
EDEITOR: In the footnote, the calculated concentrations of enzymes (beta-mannanase, phytase, and other carbohydrase) in the experimental diets need to be provided
ANSWER: Yes, the information was added (Lines: 134-136).
EDITOR: Would it be possible to indicate the concentration of substrates for beta-mannanase?
ANSWER: The measurement of β-mannans in the laboratory is challenging, and the techniques available typically rely on commercial β-mannanase enzymes. Consequently, the β-mannan concentration is more often estimated than directly measured. The estimated concentration of substrate for β-mannanase in the experimental diets was 0.528%. From the full list of ingredients used in this trial, β-mannans are present only in corn and soybean meal, both of which were consistently included across the three treatments. In the estimation, the β-mannans concentration for corn and soybean were considered at 0.14% and 1.26%, as proposed by Hsiao et al. [14]. (https://doi.org/10.1093/ps/85.8.1430). This information was added in the manuscript, as suggested (Lines: 84-86).
EDITOR: L 417: Avoid the expression of … and/or … Here, may want to rephrase the first sentence of the paragraph.
ANSWER: Yes, fixed.
EDITOR: L 463: Avoid the expression of … and/or …
ANSWER: Yes, fixed.

Reviewer 2 Report (Previous Reviewer 3)
Comments and Suggestions for Authors
I appreciate time and effort authors contributed to respond to the comments.
I am satisfied with their response and have no further comments.
Thanks.
Author Response
Dear Editor and Reviewers,
We do appreciate your consideration in reviewing the manuscript. Thank you for taking the time to evaluate it. All the suggestions have been carefully considered in the revised version of the manuscript. Our answers are listed below and an updated version of the manuscript with all the modifications (i.e., corrections and additional information inserted) highlighted in red is provided.
The surname of the author Alicia Zem Fraga was typed wrong. The correct is "Zem" instead of “Zam”.
Yours sincerely,
Gabriela Miotto Galli, main author
Ines Andretta, correspondent author
EDITOR
EDITOR: Table 1: In the footnote #2, the vitamin and trace element concentrations need to be provided as per kilogram of mixed diet. The present values appear to be to concentrations in the premix.
ANSWER: Yes, thank you so much for point out this error.
We added: 2 Premix with vitamins and minerals per kg/feed. Vitamin A: 6.400 UI/kg; vitamin B1: 0.900 mg/kg; vitamin B2: 0.0057 g/kg; vitamin K3 3.300 mg/kg; vitamin B6: 1.800 mg/kg; vitamin B12 26.00 mcg/kg; vitamin D3: 1.325 UI/kg; vitamin E: 36.10 UI/kg; folic acid 0.300 mg/kg; niacin: 0.040 g/kg; biotin 0.100 mg/kg; pantothenic acid: 0.020 g/kg. calcium 0.045 g/kg; phosphorus 0.010 g/kg; sodium 0.017 g/kg; copper 0.500 mg/kg; iron 0.500 mg/kg; iodine 0.01 mg/kg; manganese 0.250 mg/kg; zinc 0.007 g/kg.
EDEITOR: In the footnote, the calculated concentrations of enzymes (beta-mannanase, phytase, and other carbohydrase) in the experimental diets need to be provided
ANSWER: Yes, the information was added (Lines: 134-136).
EDITOR: Would it be possible to indicate the concentration of substrates for beta-mannanase?
ANSWER: The measurement of β-mannans in the laboratory is challenging, and the techniques available typically rely on commercial β-mannanase enzymes. Consequently, the β-mannan concentration is more often estimated than directly measured. The estimated concentration of substrate for β-mannanase in the experimental diets was 0.528%. From the full list of ingredients used in this trial, β-mannans are present only in corn and soybean meal, both of which were consistently included across the three treatments. In the estimation, the β-mannans concentration for corn and soybean were considered at 0.14% and 1.26%, as proposed by Hsiao et al. [14]. (https://doi.org/10.1093/ps/85.8.1430). This information was added in the manuscript, as suggested (Lines: 84-86).
EDITOR: L 417: Avoid the expression of … and/or … Here, may want to rephrase the first sentence of the paragraph.
ANSWER: Yes, fixed.
EDITOR: L 463: Avoid the expression of … and/or …
ANSWER: Yes, fixed.
This manuscript is a resubmission of an earlier submission. The following is a list of the peer review reports and author responses from that submission.
Round 1
Reviewer 1 Report
Comments and Suggestions for Authors
Review - animals-2950293
Complementary effects of β-mannanase and multi-carbohydrase complex on nutrient metabolism and gut health of growing pigs
Dear author,
The subject is within the scope of the journal, and the study scientifically sounds. The study evaluates effects whether the addition of β-mannanase alone or combined with a multi-carbohydrase complex can improve diet digestibility, nutrient, and energy metabolism, gut health using growing pigs.
They concluded that use of β-mannanase in growing pigs diet improved nutritional digestibility and nutrient and energy metabolism, gut health, and reduced manure production.
Generally, the topic is interesting. I think this manuscript could be accepted but needs some modifications and attention before the publication. I have some comments before my final decision to reject or accept it for publication.
- Please write the full name of the abbreviations in their first mentions and throughout the manuscript.
Details of major comments to the manuscript are given below:
ABSTRACT:
- Seems well written.
- Keywords: Check journal recommendations for the number of keywords used.
INTRODUCTION:
- The introduction of manuscript is diluted and semi adequate in respect to reviewing the latest literature, you need to include more information.
- The section should be amended using more recent scientific references.
- I would add a few sentences explaining what additional information your study will bring.
Materials and methods:
Seems well written. However, in several parts of the article, the sentences have not been supported by references. I need to revise and add proper references.
RESULTS:
The section seems good, however I suggest omitting all sentences with no significant effect observed.
DISCUSSION:
- In general, the production data are interesting but has not been discussed thoroughly. Although the manuscript's discussion seems somewhat adequate, I recommend enhancing the discussion section by emphasizing your direct observations. I would suggest making every effort to provide more detailed explanations of the results without necessarily comparing them to other studies.
- Furthermore, the discussion should incorporate additional arguments from other studies or results found in the literature.
- Additionally, the findings of this study should be linked to the suggested pathways or implications of the treatments utilized on the different variables evaluated.
CONCLUSION:
- Need to make it clear and avoid repetition of what you have already written previously.
- Please avoid using abbreviations in this section.
REFERENCES:
- Should be updated as some references are old and suggest using recent 10-year references. Can you confirm this with references that are more recent?
TABLES: Seems well.
FIGURES: Seems well
Reviewer 2 Report
Comments and Suggestions for Authors
This paper presents a study of addition of BM alone or with MCC to access the beneficials of these additions in growing pigs. The trail was well designed and carried out, data were rationally analyzed. I believe this paper would add more information to readers on this point. The following points must be concerned before publication.
1. abstract: what type of feed was used should be mentioned(corn-soybean meal).
2. material and methods
enzymes: more detail information is needed, such as how many activity units?
3. Results:
3.1 can be removed to material and methods
in tab7, value of c-protein and transferrin in serum among groups is exactly the same?
Please add n in each table.
4. Discussion:
As your hypothesis is that “whether the addition of β-man-nanase alone or combined with the multi-carbohydrase complex can act additively to improve diet digestibility, nutrient, and energy metabolism, as well as gut health in growing pigs”, the discussion mentioned little on this point. Please give more explanation in this point, since you did not get any additive effect in BM+MCC
Intestinal morphometry indexes checked in this study are need further discussed, since they are varied in different sections.
Reviewer 3 Report
Comments and Suggestions for Authors
Thank you for submitting your manuscript entitled "Complementary effects of β-mannanase and multi-carbohydrase complex on nutrient metabolism and gut health of growing pigs" to "Animals".
After careful consideration, I believe that your manuscript has the potential to make a contribution to the area. However, before it can be considered for publication, significant revisions are required.
Minor edits:
L73+ While the manuscript mentions the enzymes used, it lacks detailed specificity regarding the item numbers, batch numbers, or the protocols of enzyme production. Such details are crucial for reproducibility and understanding the exact conditions under which the experiments were conducted.
L79 The manuscript does not provide a clear justification for the number of animals used in the study. A rationale for choosing 24 pigs, including any statistical power analysis, would strengthen the experimental design section and ensure the study is adequately powered to detect significant effects.
L144 Reference to standard protocols or guidelines, especially regarding feed retention rate and fecal moisture content measurements, is not explicitly mentioned. Clarification on whether section 2.3.2 follows a standard protocol and the inclusion of references or justifications would enhance the credibility and reproducibility of these methods.
L151 The manuscript could benefit from a graphical timeline of the experimental procedures. Such a visual aid would help readers quickly grasp the study's design and timeline, improving the manuscript's overall clarity and accessibility.
L165 The manuscript lacks specific details regarding the test kits used for various measurements, such as acute phase proteins and fecal biomarkers.
L192 Room ambient temperature and average humidity are not results of experiment. Such environmental parameters are typically considered part of the materials and methods, as they relate to the experimental conditions rather than the results of the study.
[Results] section: While the manuscript includes tables, the suggestion for graphical representation of results indicates a potential flaw in the presentation. Incorporating graphs could provide a more intuitive understanding of the data and reveal trends or patterns not immediately apparent in tabular form.
[Discussion] section:
1) Manuscript might not sufficiently emphasize the study's novelty or its contribution to new knowledge within the field. Strengthening the argument for how this study adds unique insights could elevate the manuscript's impact.
2) Manuscript may not adequately discuss its own limitations. Acknowledging limitations in the experimental design, sample size, or other aspects of the study is crucial for providing a balanced and honest interpretation of the results.
3) While the manuscript concludes with the effects of enzyme supplementation, it could further elaborate on specific questions or challenges that emerged from the study, setting a clearer direction for future research in this area.
Overall:
The manuscript might benefit from deeper statistical analyses, such as interaction effects between different treatments or time-series analyses, to fully explore the data. Advanced statistical methods could reveal more nuanced insights into how the enzyme treatments affect pigs over the course of the experiment.
While the manuscript discusses the effects of enzyme supplementation, it may lack a detailed exploration of the biological or chemical mechanisms underlying these effects. Providing a hypothesis or discussing potential mechanisms could enrich the discussion and suggest pathways for future experimental work.
The manuscript overlooks the impact of enzyme supplementation on the gut microbiota composition, which is a significant area of interest in animal nutrition. Integrating microbiota analysis could provide a more comprehensive understanding of how these enzymes influence gut health.
The manuscript does not fully address how the composition of the pigs' diet (beyond the addition of enzymes) could influence the efficacy of the enzyme supplementation. Discussing or controlling for diet composition could help isolate the effects of the enzymes.
The study's duration does not capture long-term effects of enzyme supplementation on growth performance, nutrient metabolism, and gut health. Discussing the potential for longer-term studies or the sustainability of enzyme use in pig diets would add depth to the conclusions. Addressing this point in "limitations" would raise credibitlity of the manuscript.